# Anxiety classification in virtual reality using biosensors: A mini scoping review

**Deniz Mevlevioğlu**[1]*, **Sabin Tabirca**[1,2], **David Murphy**[1]

**1** Department of Computer Science, University College Cork, Cork, Ireland, **2** Faculty of Mathematics and Informatics, Transylvania University of Brasov, Brașov Romania

* d.mevlevioglu@cs.ucc.ie

## Abstract

### Background

Anxiety prediction can be used for enhancing Virtual Reality applications. We aimed to assess the evidence on whether anxiety can be accurately classified in Virtual Reality.

### Methods

We conducted a scoping review using Scopus, Web of Science, IEEE Xplore, and ACM Digital Library as data sources. Our search included studies from 2010 to 2022. Our inclusion criteria were peer-reviewed studies which take place in a Virtual Reality environment and assess the user's anxiety using machine learning classification models and biosensors.

### Results

1749 records were identified and out of these, 11 (n = 237) studies were selected. Studies had varying numbers of outputs, from two outputs to eleven. Accuracy of anxiety classification for two-output models ranged from 75% to 96.4%; accuracy for three-output models ranged from 67.5% to 96.3%; accuracy for four-output models ranged from 38.8% to 86.3%. The most commonly used measures were electrodermal activity and heart rate.

### Conclusion

Results show that it is possible to create high-accuracy models to determine anxiety in real time. However, it should be noted that there is a lack of standardisation when it comes to defining ground truth for anxiety, making these results difficult to interpret. Additionally, many of these studies included small samples consisting of mostly students, which may bias the results. Future studies should be very careful in defining anxiety and aim for a more inclusive and larger sample. It is also important to research the application of the classification by conducting longitudinal studies.

**Data Availability Statement:** All relevant data are within the paper and its Supporting information files.

**Funding:** DMe Received a grant from Science Foundation Ireland with the number 18/CRT/6222

https://www.sfi.ie/ The funders had no role in study design, data collection and analysis, decision to publish, or preparation of the manuscript.

**Competing interests:** NO authors have competing interests.

## Introduction

Virtual Reality (VR) has become more and more popular by the day, while the use of VR for medicinal and therapeutical reasons is almost commonplace. While VR can make valuable contributions to the health and well-being of society, it is important to research and monitor the effects it has on individuals.

One such area in that VR has been evidenced to be very effective in fighting anxiety disorders [1]. As more proof of the efficiency of VR therapy unearths, the study into feedback receival within Virtual Environments (VE) becomes more important. One such feedback is how anxious the user feels within such an environment.

Real-time anxiety classification within VR enables a range of possibilities for VR therapy, supporting decision-making in VR and providing feedback on the effectiveness of the intervention. It, unfortunately, comes with a set of challenges, such as the bias introduced by self-assessing anxiety [2], not breaking the immersion in VR and the technical difficulty of systems that measure physiological signals while the user is mobile [3].

The current study is a mini-scoping review using PRISMA-ScR guidelines with the topic of anxiety classification in VR using biosensors [4].

### Definition of anxiety

APA defines anxiety as an emotion response characterised by negative feelings of worrying and physical symptoms such as higher blood pressure, sweating, dizziness and elevated heart rate [5]. Stress is characterised by a similar emotional and physiological response, however, it is usually caused by an external factor as opposed to a prolonged and internal feeling [6]. Fear is sometimes used interchangeably with anxiety, however, it differs from anxiety for being short-lived and responding to an identifiable threat [6].

This study acknowledges the differences between stress, anxiety and fear. However, due to the similar physiological response of these emotions, they are all included in this study for the sake of identifying physiological systems that can classify and cluster these responses into their respective emotion. These studies generally have a common methodology, thus, it is beneficial to investigate all of them.

### Stress response

Stress is a useful feeling that can trigger a fight-or-flight response when a person faces danger [7]. However, if this response is regularly activated without the presence of a real threat, it might become detrimental to the person's health [8].

Physiological outcomes of anxiety depend on several factors such as the person's age and genetics; as such, these vary from person to person. The short-term response to stress can lead to increased sweating, pupil dilation, increased heart rate and respiratory rate [9]. The long-term response to stress can lead to increased synthesis of cortisol [10].

In the existence of a stressor, two systems respond to make sure that the body deals with the stressor efficiently. The first response is from the sympathetic adrenomedullary system (SAM) and it is quick and short-lasting [10]. SAM axis works by neurally instigating responses in specific organs with the help of adrenaline produced by stimulation of the hypothalamus [11] This results in increased sweating, pupil dilation, increased heart rate and breathing rate [9].

The second phase is the response from the Hypothalamic Pituitary adrenal (HPA) axis, which is slower and long-lasting [10]. The HPA axis is instigated by the hypothalamus and responds through hormones [11] and is signified by the synthesis of cortisol.

## Physiological and behavioural measures

Due to their physiological response, the anxiety level is strongly and positively correlated with both skin conductivity level (SCL) and heart rate (HR); resulting in these measures being among the most popular for anxiety classification [3]. Other physiological measures that are correlated with anxiety include skin temperature, blood volume pressure, electrical brain activity and electrical muscle response [3]. Compared to heart rate and skin conductance, the interaction between these measures and anxiety is more complicated, however, they are evidenced to be useful measures.

Anxiety also results in some behavioural responses, which can be detected through behavioural measures such as head movement, eye movement and respiration. Evidence suggests that this information also has a relation with anxiety, making them useful metrics for classification [12, 13].

Salivary cortisol levels are positively correlated with anxiety and stress [3, 10], however, the methods for determining levels of cortisol in saliva are generally invasive which might not be optimal for an immersive VR experience.

Prior studies made use of such measures and machine learning models to classify predefined levels of anxiety, such as predicting if the person is currently experiencing relaxation, mild anxiety or severe anxiety [13].

Over the last decades, there has been increasing research in classifying anxiety using machine learning and sensors [3]. Unfortunately, many of the systems used to detect anxiety using sensors limit the mobility of the user [13], which renders them unsuitable for VR treatments as movement plays a very important role in presence within VR [14, 15]. This gap indicates a need for investigating objective measures using physiological and behavioural information that is suitable for use in VR.

## Application areas

VR is commonly used for therapy, meditation and medicinal purposes [1, 16]. Systems that incorporate wearable sensors are not commercially widespread, however, the research in this area is currently extending due to its many areas of application [17]. Using VR for health care can increase cost efficiency and provide support when in-vivo options are difficult to reach, and it can also help reduce the load of health care systems [18, 19], therefore, VR treatment methods for psychological disorders are becoming increasingly popular while the scientific evidence points to similar effectiveness when compared to in-vivo treatment methods [16, 20].

As VR therapy has clear benefits of cost-efficiency and availability, it is important for the wellbeing of society to research it as a medium [1, 18]. Although it might not be accessible for everyone at the moment due to problems with adjustment to the virtual world [21], possible visual difficulties for some users [22], and motion sickness [23, 24], it is still an important addition to current therapy methods [25]. For the improvement of VR therapy, feedback from the patient is very important. While the most direct approach to this is asking the user, this can either be disruptive or unhelpful. Also, it is difficult to get self-report measures immediately, and it can be difficult to self-assess retrospectively as the user might have already forgotten how they were feeling at the time. This is why it is crucial to have a real-time solution. Physiological signals using wearable sensors provide a way of receiving feedback at the moment, objectively, and without interruptions to the experience.

One important application of anxiety detection in VR is its use in conjunction with Virtual Reality Exposure Therapy (VRET) [26]. VRET is a therapy method to treat certain anxiety disorders by allowing the user to face their fears in the safety of the provided virtual environments [27]. As every person is unique and responds to therapy differently, we argue that it is

important to predict the anxiety level of the user to make better-informed decisions on the exposure adjustment.]

In addition to its medicinal and therapeutical applications, real-time anxiety classification in VR is useful for any type of application that uses feedback. How users are feeling at the moment can be used to adjust the content, make improvements to software and change settings. One example of its use is providing insight into user state while performing challenging tasks [28]. It is also interesting to look at it for entertainment purposes, using the classification of anxiety to adjust difficulty/atmosphere in immersive VR games [29].

### Research questions

This study sets out to answer the following research questions:

- RQ1: What are the most common measures for anxiety classification, and what are their advantages and disadvantages?

- RQ2: What are the current ground truth establishment methods used to classify anxiety, and what are their advantages and disadvantages?

- RQ3: What are the classification methods most commonly used and what are their advantages and disadvantages?

- RQ4: What are the gaps in the literature relating to real-time anxiety prediction in Virtual Reality?

## Materials and methods

Inclusion and exclusion criteria can be viewed in Table 1.

The following databases were searched for eligible studies:

- IEEE Xplore

- ACM Digital Library

- Scopus

- Web of Science

The following strategy was used to search each database:

Title, abstract or topic including "Virtual Reality" or "VR" and "anxiety" or "stress" and "recogn*" or "detect*" or "predict*" or "classif*" or "discriminat*" or "identif*", filtered to only include publications published between 2010 and 2022.

Supplementary sources were scanned using references from identified papers, past reviews with similar titles and Google Scholar.

**Table 1. Inclusion and exclusion criteria.**

| Inclusion Criteria | Exclusion Criteria |
| --- | --- |
| The study was conducted in a VR environment | The anxiety classification is not done in a VR environment |
| The experiment examines user anxiety in real-time using classification models | The anxiety classification cannot be applied in real-time |
| The paper is peer-reviewed short or full paper | The study is not a full or short paper or not peer-reviewed |
| The study was published in or after 2010 | The study was published before 2010 |

After records were identified, duplicates were removed. The resulting records' titles and abstracts were then screened by a single reviewer for relevancy. The relevant papers were fully examined for eligibility.

The review was not registered and a protocol was not prepared. The completed PRISMA-ScR checklist can be found in S2 File.

A data extraction form was created and pilot tested on five studies. The data were extracted from the studies using the data extraction form. The full data extraction form can be viewed in S3 File. The extracted data can be viewed in S1 Table.

The following are the items in the data extraction form:

- DI1: Reference (First author / Year / Citation)

- DI2: Sample size

- DI3: Type(s) of physiological or behavioural measurements (heart rate, eye movement, etc.) used

- DI4: Ground truth establishment method used

- DI5: Classification model that achieved the highest accuracy

- DI6: Numbers of outputs for the classification

- DI7: Quality score

- DI8: Highest accuracy

The quality assessment scale was modified from the [30] qualitative study checklist and pilot tested on five studies. It has eight items, with possible answers of yes (1 point), no (0 points) and partially (0.5 points). The possible quality score ranges from 0–8. The full checklist can be viewed in S1 File.

## Results

A summary of the search results is presented in Fig 1. There were no papers identified through supplementary sources.

Sixteen studies were excluded because they were found to be ineligible upon further investigation [31–43]. Exclusion Criteria 2 (classification does not take place in a VR environment) accounted for nine of these exclusions [32–35, 38, 40–42, 44], and Exclusion Criteria 3 (no classification that can be applied in real-time) accounted for seven [31, 36, 37, 39, 43, 45, 46]. The studies by Handouzi et al. [33–35] state that their studies take place in a VR environment but the content is displayed on a 2D screen while the participant is asked not to move their limbs, so these papers were deemed ineligible due to not being applicable to VR.

The detailed quality scores of each study can be viewed in Fig 2.

A summary of the data extraction from eligible papers is presented in Table 2. Most of the studies achieved quality scores over 6, with the only exception being the study by [48]. When interpreting the table of results, it should be kept in mind that the accuracy of the models cannot be compared directly due to differences between studies in terms of what they classify as anxiety.

### Physiological and behavioural measures

There was a large variety of measures used between studies. Behavioural measures used were respiration (RESP) [48, 56, 58], head movement [54] and eye movement [53]. Physiological measures used by high-quality studies were electrodermal activity (EDA) [50, 51, 55,

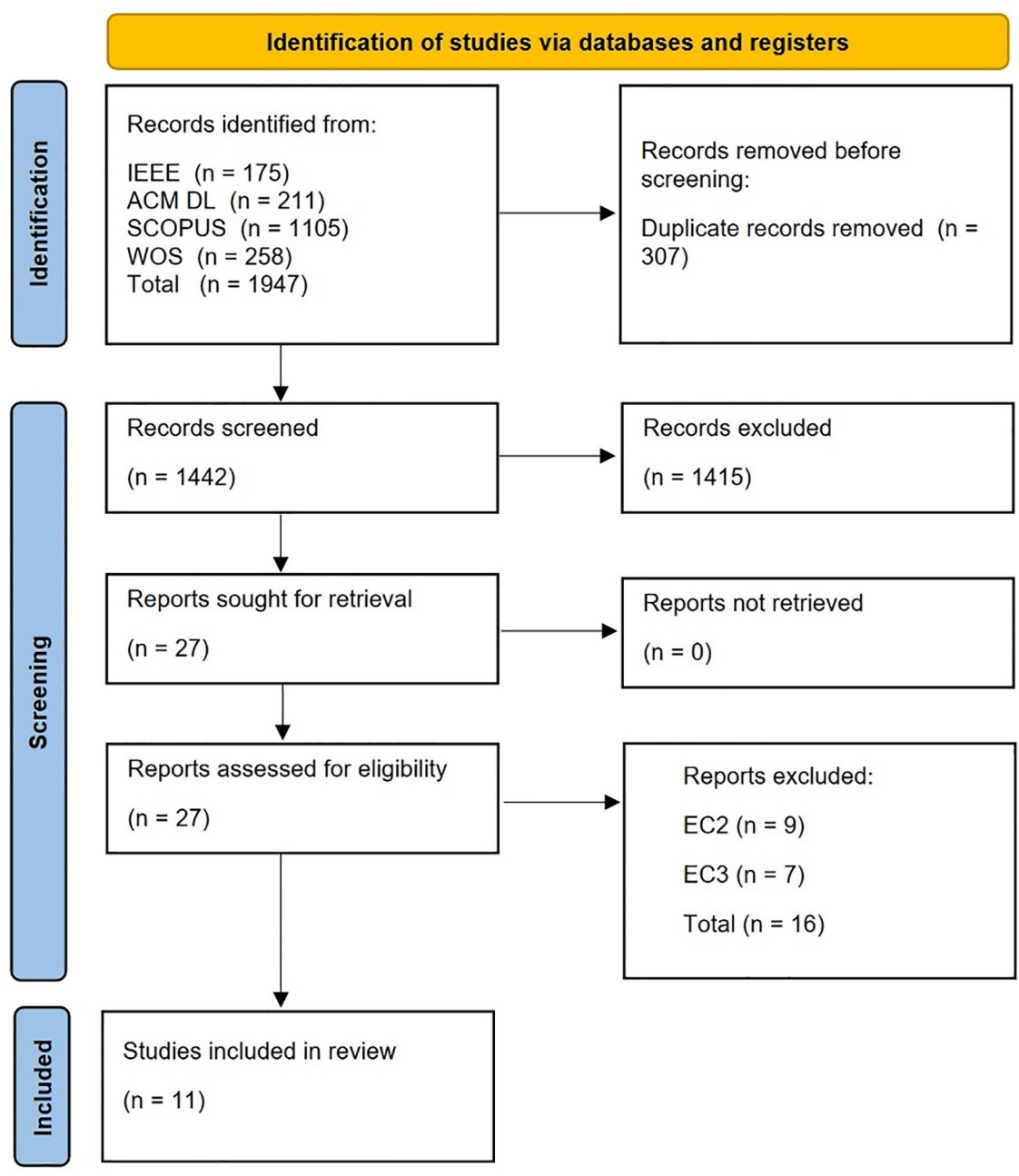

**Fig 1. The search results [4].**

57, 58], heart rate (HR) and heart rate variability (HRV) measured by electrocardiography (ECG) [48, 49, 52, 56, 58] and pulse rate (PRV) and pulse rate variability (PRV) measured by photoplethysmography (PPG) [51, 52, 55], brain electrical activity measured by an electroencephalogram (EEG) [48, 50, 53, 58] and skin temperature (SKT) [51, 57]. [48] also stated that they used electromyography (EMG), temperature (TMP) and accelerometer (ACC). However, there was no information provided about these sensors, and how and why they were used.

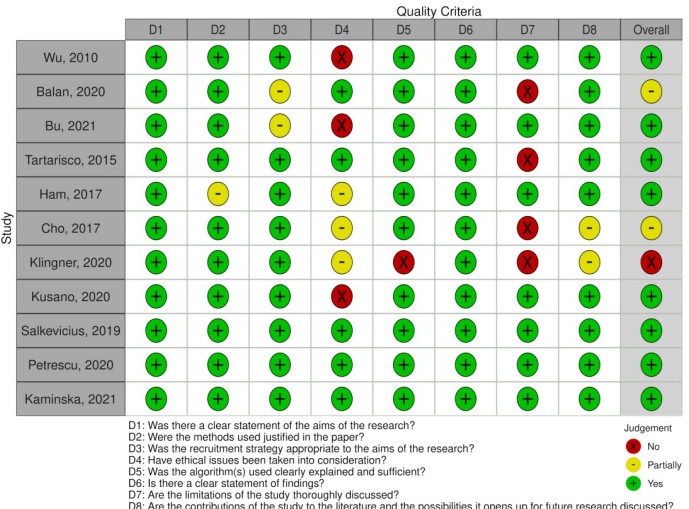

**Fig 2. Quality scores of each study [47].**

Some of these studies combined behavioural and physiological measurements [48, 53, 56, 58] while others focused on physiological measures alone. [54] used only head movement to classify anxiety levels.

As stated before, one of the biggest challenges in deciding the measures is getting objective and high-quality information without causing discomfort in the VR environment, which can

**Table 2. Previous research on real-time anxiety prediction in VR.**

| Ref | Sample size | Measures | Ground truth | Highest Accuracy Model | Number of Outputs | Quality score | Highest accuracy achieved |
|-----|-------------|----------|--------------|------------------------|-------------------|---------------|---------------------------|
| [49] | 8 | HRV (ECG) | VR Content | LSTM | 2, 3, 4 | 6.5 | 90.5% (2-Level), 67.5% (3-Level), 58.8% (4-Level), 30s |
| [50] | 4 | PPG, EDA, EEG | SUDS | ANN | 11, 4, 2 | 6.5 | 78.3% (2-Level), 38.8% (4-Level), 26.5% (11-Level) |
| [51] | 12 | PPG, EDA, SKT | VR Content, STAI-Y1 | K-ELM | 3 | 6 | 96.3% |
| [52] | 6 | ECG, PPG | VR Content | LDA | 3 | 6.5 | 79% |
| [53] | 28 | EEG, EMDR | SCWT | MLP | 2 | 8 | 96.42% |
| [48] | 1 | ECG, EDA, EMG, EEG, RESP, ACC, TMP | VR Content | SVM | 2 | 4.5 | 82% |
| [54] | 73 | Head movement | Card Sort, COMP-TRACK, Movies | SVM | 2 | 7 | 75% |
| [55] | 7 | EDA, PPG | SUDS | CNN | 2, 3 | 8 | 94.3% (2-Level), 92.4% (3-level) |
| [56] | 20 | ECG, RESP | STAI-Y1 | Neuro-fuzzy | 4 | 7 | 83% |
| [57] | 30 | BVP, EDA, SKT | VR Content, SUDS | SVM | 4 | 8 | 86.3% |
| [58] | 19 | EDA, RESP, ECG, EEG | SCWT | SVM | 3 | 7 | 84% |

Note: ACC: Accelerometer, BVP: Blood Volume Pressure, ECG: Electrocardiogram, EDA: Electrodermal Activity, EEG: Electroencephalogram, EMDR: Eye Movement Desensitization and Reprocessing, EMG: Electromyogram, HRV: Heart rate variability, PPG: Photoplethysmogram, RESP: respiration, SKT: Skin temperature, TMP: Temperature, SCWT: Stroop Colour-Word Task, SUDS: Subjective Units of Distress Scale, STAI: State-Trait Anxiety Inventory, LSTM: Long Short Term Memory, ANN: Artificial Neural Network, K-ELM: Kernel Extreme Learning Machine, LDA: Linear Discriminant Analysis, MLP: Multi-layer Perceptron, SVM: Support Vector Machine, CNN: Convolutional Neural Network.

have a toll on immersion. However, it is hard to interpret physiological information from bio-sensors, due to the difficulty of producing noise-free data and the methods of removing noise in real-time being limited. As a result of this, many studies [48, 50–53, 55–58] chose to use multi-modal systems incorporating a range of measures to compensate for the accuracy loss in the case of receiving low-quality signal from any one sensor. It can be argued, however, that increasing the number of biosensors can decrease the sense of presence within VR and that sense of presence is important for VR therapy [20]. Thus, some studies chose to limit the number of measures to one or two measures [49, 52–56].

## Ground truth

Ground truth refers to the point of reference used to differentiate between anxiety and no-anxiety states within the experiments. Establishing a ground truth of anxiety is a difficult procedure due to the difficulty of assessing anxiety itself [59]. In literature, different approaches have been taken in the establishment of ground truth.

Some studies use self-report measures or questionnaires such as subjective units of distress scale (SUDS) or scale of trait anxiety inventory (STAI) [50, 51, 56, 57]. This reduces the ambiguity of what is being detected in the environment, which means there are fewer assumptions. However, if used alone, this method can lead to bias due to personal differences between participants and the difficulty of arbitrarily describing one's own anxiety levels.

Other studies use well-established stress-inducing cognitive tasks such as Stroop Colour-Word Task (SCWT) or card sort task [53, 54, 58]. These types of cognitive tasks have been widely used in literature and have been validated to cause certain stress reactions. Therefore, some studies use them to objectively label levels of anxiety, reducing the subjectivity of the measure.

Some studies preferred to make their own relaxing and anxious VR environments and validated these to be used as ground truth in their studies [48, 49, 52]. The advantage of this approach is its applicability to its use case. However, studies that only use this approach introduce a risk of bias to their results due to the difficulty of validating their results.

## Classification methods

Studies included in this review tried different features and models to come up with the highest accuracy solution (Table 2). Not all of the included studies aimed for real-time classification, so for some studies, high performance is not a priority. Given the nature of the features, some studies succeeded in reaching optimal accuracy using relatively simple models such as Linear Discriminant Analysis (LDA) [52] and Support Vector Machines (SVM) [57, 58]. [55] found better results using a Convolutional Neural Network (CNN). Some studies found high-accuracy results without compromising performance by using feedforward neural networks such as Kernel Extreme Learning Machine (K-ELM) [51] and Neuro-fuzzy systems [56]. However, it should be kept in mind that there are other effects in play such as biosensors and stressors, and the accuracy of the described models can only be evaluated within their studies.

Features used in the evaluated studies heavily depend on the type of measures they were using. They are commonly grouped into time-domain, frequency domain and non-linear measures. Common statistical features used across most signals were mean, average, minimum, maximum, standard deviation and variance [51–53, 55, 57, 58]. Common features used relating to ECG and PPG were Inter-beat Intervals (IBI) and Normal-to-Normal (NN) average and standard deviations in the time domain, and low and high-frequency component averages in the frequency domain [51, 52, 55, 58]. For EDA or GSR, these included event-related responses using peak detection such as number of peaks, peak frequency and average peak amplitude

and maximum peak amplitude [51, 55, 57]. Approximate entropy and sample entropy were non-linear features used to quantify regularity and predictability [51, 52].

Preparation for physiological and behavioural data vary. For information such as ECG signals and EEG signals, it is very common to use bandpass filters to avoid using artifacts introduced from movement and electrical interference [51–53, 55, 57, 58]. Normalisation was generally applied in studies that use ANNs and SVMs [55, 57, 58]. Additionally, Petrescu et al. [55] applied down-sampling because their EDA signal's sampling rate was higher than that imposed by its bandwidth. For EEG, the data was split into separate wavelengths of delta, theta, alpha, beta, low beta, high beta and gamma [53, 58]. Wu et al. [58] focussed specifically on alpha, beta and theta. Independent Component Analysis (ICA) was used by Kamińska et al. [53] for processing EEG data to remove unwanted noise.

The studies used various feature selection methods to simplify the models after extracting a high number of potential features. Most of the studies in this review used different methods, including Sequential Forward Selection (SFS) [58], Pearson correlation [55], ReliefF and Davies-Bouldin cluster evaluation index [56].

Time windows were relatively small across most studies. Eight of the studies ranged from 3s [53, 58] to 30s [50] window durations. The study by Bu et al. [49] featured time windows of five minutes. Only three studies used windows longer than 6s [50, 54, 57]. Two studies did not specify their time window [48, 52]. As the time windows get bigger, it is easier to make accurate estimations. However, the resulting models might end up being less usable by making the real-time estimations too vague. So, a good balance between accuracy and using smaller time windows is important.

The number of outputs that were classified in these models ranged from two to eleven across all studies, with the most common being two. Many studies used two-output models (no anxiety vs anxiety) because it can be difficult to define and detect increasing levels of anxiety, increasing the challenge of labelling data for classification. However, some studies found success with increased levels of outputs, giving more detailed information regarding the anxiety state of the user [51, 55, 57].

It is hard to determine here what the "acceptable" accuracy is due to the differences and difficulties in defining different anxiety states. Across the studies reviewed, accuracy for two output models ranged between 75% [54] and 96.4% [53]; accuracy for three output models ranged between 67.5% [49] and 96.3% [51]; accuracy for four output models ranged between 38.8% [50] and 86.3% [57]. In the study by [50], the accuracy results are compared to the chance of getting the correct results when making a random guess (50% for 2-choice, 25% for 4-choice), however, it is hard to determine how much practical use can come out from such a comparison. Higher accuracy is desirable as low-accuracy models will not be reliable, however, the practical use of the model is important as well. If what is being predicted does not prove useful in a VR application, the accuracy of the prediction means very little. Furthermore, user comfort must not be compromised when increasing classification accuracy.

## Gaps in the literature

It is difficult to ascertain whether there are enough studies in this specific area yet to conclude the usefulness of anxiety classification systems. The up-to-date studies, albeit their problems, show promise for the future. However, there is a big lack of in-the-wild solutions, meaning that most studies were conducted in labs under controlled conditions. Furthermore, in many of the studies, data collection is only done once and the results are validated in past data, and not in real-time. Although varying between studies, the accuracy of classification models approaches useful levels. While these results are useful in their contained studies and VR

environments, few solutions have been tested across different environments and samples. This makes the generalisability very difficult to assess. Furthermore, there are gaps in the application of these systems to clinical trials even though many of them are developed to aid therapy.

## Discussion

There are many different measures used in anxiety classification in VR, behavioural or physiological. Many studies used heart rate and electrodermal activity to detect anxiety [48, 50, 51, 55, 58]. From the behavioural measures, respiration was the most prevalent [48, 56, 58]. Many high-quality studies were able to achieve promising results by combining several measures to strengthen the reliability of their system [53, 56, 58]. This might suggest that it is a good idea for future studies to use at least two different measures while classifying anxiety in VR.

Current literature has no agreement regarding important issues that lie in the foundation of the problem of anxiety classification in VR. One of the key issues is the definition of anxiety and relaxation itself. It is understandable that studies measure different levels of anxiety, however, these levels are not linear or easily distinguishable between the studies. What one study describes as the absence of anxiety, another might describe as mild anxiety. As such, it is difficult to answer whether it is possible to accurately classify anxiety in VR. Certainly, there were many promising results and high-quality research when it came to achieving high internal accuracy [53, 55, 57]. However, the question remains whether it is possible to claim that the anxiety measured here is generalisable and valid. For this to be possible, there needs to be some unification of accepted standards. Thus, working on generalisable standards for anxiety measures must be central for future studies.

### Limitations

The studies included in this review had several important limitations. Some studies poorly described their samples and methods [48, 49] which made it difficult to extract meaningful information from these studies. Many of the studies had sample sizes below 10, which makes interpretation of the results difficult [49, 50, 52, 55]. Furthermore, most studies were done in universities and had a young population, with some samples' mean age being under 30 [51, 52, 57]. This is problematic due to two reasons. First, the results can only be interpreted for younger people, which means that the prediction might not be accurate when administered to the general population. Second, it is hard to determine the problems in the system when used by older people. Some studies make the inference that their results show sufficient success in predicting accuracy with little to no evidence [48, 50].

This study used single screening, which, based on research might introduce some errors in abstract screening and increases the risk of missing eligible studies [60, 61]. Also, the study quality criteria creation, rating of study quality and data extraction were conducted by a single reviewer and later verified by a second reviewer. This might have introduced a risk of bias to our study. However, we believe that these limitations would not have majorly impacted the conclusion of the study.

### Future directions

This field is still relatively new but it enjoys consistent attention and progresses steadily. Although the direction of research here may be promising, there is a need for a lot more research before it can be used in daily life.

First of all, there need to be a lot more studies on best practices when it comes to establishing ground truth. Studies that test ground truth detection methods over different applications

and settings are vital for the validation of anxiety levels. Without this, it is very difficult to ascertain what is actually being measured.

Second, a lot of the studies in the literature thus far focus on laboratory settings and are contained within one application. There is close to no reproducing of the results, which leads to questions about the validity of the results when it comes to wider applications. The variability in the ground truth, VR content and measures used make it difficult to directly compare results. Thus, it is important to reproduce previous work using the same methods on different samples.

Another area that needs attention is the application of these systems to clinical trials to validate whether it is useful in their application areas. In-the-wild testing is required to further validate the accuracy of the systems, as well as their usability.

## Conclusion

This review shows that it is possible to consistently measure the anxiety state of the user for the purpose of research. The use of anxiety classification within VR enables developers to further tailor experiences to the user, possibly increasing the effectiveness of therapeutical or entertainment VR applications. It is important for future research to study the benefits of anxiety classification to such applications, for instance, the effectiveness of VRET enhanced by real-time anxiety classification as opposed to traditional systems. To study such an effect, longitudinal studies would be necessary.

Based on this review, it is important for future studies to be very precise with the methods they use to be able to derive meaning from the results. The focus needs to be concentrated on establishing methods that can be used through different systems and applications.

## Supporting information

**S1 Table. Data extraction table.**
(XLSX)

**S1 File. Quality criteria.**
(PDF)

**S2 File. PRISMA checklist.**
(PDF)

**S3 File. Data extraction form.**
(DOCX)

## Author Contributions

**Conceptualization:** Deniz Mevlevioğlu, Sabin Tabirca, David Murphy.

**Data curation:** Deniz Mevlevioğlu.

**Formal analysis:** Deniz Mevlevioğlu.

**Methodology:** Deniz Mevlevioğlu.

**Project administration:** Deniz Mevlevioğlu, Sabin Tabirca, David Murphy.

**Supervision:** Sabin Tabirca, David Murphy.

**Validation:** Sabin Tabirca, David Murphy.

**Writing – original draft:** Deniz Mevlevioğlu.

**Writing – review & editing:** Sabin Tabirca, David Murphy.

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
