## [Decision Letter · Decision Letter 0]

16 Mar 2023

PONE-D-23-04739Anxiety Classification in Virtual Reality Using Biosensors: A Mini Systematic ReviewPLOS ONE

Dear Dr. Mevlevioğlu,

Thank you for submitting your manuscript to PLOS ONE. After careful consideration, we feel that it has merit but does not fully meet PLOS ONE’s publication criteria as it currently stands. Therefore, we invite you to submit a revised version of the manuscript that addresses the points raised during the review process.

We look forward to receiving your revised manuscript.

Kind regards,

Humaira Nisar

Academic Editor

PLOS ONE

Journal Requirements:

"This publication has emanated from research supported in part by a Grant from 

Science Foundation Ireland under Grant number 18/CRT/6222. "

"DMe Received a grant from Science Foundation Ireland with the number 18/CRT/6222

https://www.sfi.ie/

The funders had no role in

study design, data collection and analysis,

decision to publish, or preparation of the

manuscript."

Reviewers' comments:

Reviewer's Responses to Questions

**Comments to the Author**

1. Is the manuscript technically sound, and do the data support the conclusions?

Reviewer #1: No

Reviewer #2: Yes

Reviewer #3: Partly

2. Has the statistical analysis been performed appropriately and rigorously? 

Reviewer #1: N/A

Reviewer #2: N/A

Reviewer #3: Yes

3. Have the authors made all data underlying the findings in their manuscript fully available?

Reviewer #1: Yes

Reviewer #2: Yes

Reviewer #3: Yes

4. Is the manuscript presented in an intelligible fashion and written in standard English?

Reviewer #1: Yes

Reviewer #2: Yes

Reviewer #3: Yes

5. Review Comments to the Author

Reviewer #1: Thank you for providing the opportunity to review the paper “Anxiety classification in virtual reality using biosensors: a mini systematic review”. The paper does not provide sufficient work to be included for the publication in top tier journal PLOSOne. I am not recommending this paper for publication. My over all comments are below.

The authors have not provided sufficient details of about the anxiety, VR and biosensors in the introduction part. They should expand their introduction providing more explanation of these terms which makes the basis of their work. Building on the knowledge of anxiety (or stress), VR and Biosensors, the importance of anxiety classification should be presented. The authors describes very little about its applications. More applications should be provided as evidence and their limitations should be included in the review.

The authors tried to establish their research questions but reading ahead it seems that they don’t answered their RQ and the review doesn’t look like a systemic approach. They should stick to their own RQs and provide detail answers in the light of published literature.

Inlusion/exculsion criteria should be presented in a simple table.

No details of the classification’s methods are provided. Mentioning only the name classifiers does not provide any useful information to the researchers in the field.

The discussion is just a summery of what they have presented earlier and not a real discussion over the problem of topic.

Over all the paper thoroughly needs to be revised and expand on the topic from scratch.

Reviewer #2: The Authors present a review on anxiety classification in virtual reality scenarios. Virtual reality has become a very widely used technology in many areas, including healthcare or training in dangerous professions. Research shows many advantages of this solution and emphasizes its effectiveness. The limitations of this technology should also be noted. Researchers point out several significant side effects of using virtual reality. One of them is stress and anxiety, which can be the result of long-term exposure to a stressor. It is known that the measurement of these symptoms is not simple, especially in real time when exposed to intervention in a virtual environment. Therefore, the Reviewer would like to thank the Authors for conducting such interesting secondary research. This article is quite well-prepared and conveys important knowledge; however, there are some issues that need to be revised.

GENERAL STRENGTHS

My knowledge in the field of biosensors is limited; however, because of my interest in modern technologies (including virtual reality), I consider this research to be very important in the context of the development of this technology. Therefore, I believe that the article presents important aspects related to the evaluation of side effects of therapy in a virtual environment. This is especially relevant in my field of clinical research. Content-wise, I have no complaints.

GENERAL WEAKNESSES

My expertise in the field of research design allows me to draw attention to one aspect. I recommend the Authors to change the study design. From my point of view, this study cannot be called a "systematic review". Please find below the most important comments listed point by point that explain my recommendation.

1. In general, systematic reviews (SR) are intended to summarize knowledge on a particular topic, to provide specific practical guidelines for clinicians, and recommendations for future research. It is important that such conclusions can be drawn in relation to a specific population.

2. The research question is usually very precise and allows to answer a specific clinical problem. Most of the times, a systematic review decides on the effectiveness of a therapy or compares different approaches to therapy. Meanwhile, the research questions posed by the Authors are precise; there are many of them, but they do not resolve the clinical problem.

3. One of the most important elements of SR is the creation of the PICO and the inclusion and exclusion criteria. These elements are intended to ensure precision throughout the literature review process.

4. SR requires a very precise approach. The guidelines clearly show the need to register the protocol a priori, or the participation of two independent Reviewers during the screening of abstracts, screening of full texts, and the qualitative assessment of the included articles. All these assumptions are intended to increase the relevance, reliability and objectivity of the review.

In conclusion, the Reviewer greatly appreciates the effort put into the review by the authors. Following the steps of the CONSORT guidelines is very engaging and demanding. The recommendations are as follows:

a) Change the name of the study design to “literature review” or “scoping review”; however, the Reviewer believes that "scoping review" would be a better option, as the study is based on many research questions and has an exploratory nature.

b) It is recommended to delete the PRISMA checklist, the Authors should replace it with the PRISMA-ScR checklist.

MINOR CONCERNS

1. Please consider presenting the results of the qualitative assessment in a graphical form, e.g., in a form similar to the RoB-2 figure.

2. Please try to relate the results to the qualitative assessment of the included articles.

3. In the discussion, please separate the sub-sections "Limitations" and "Recommendations for further research”. These are the areas from which the most can be drawn when designing future research. Therefore, they should be clearly highlighted.

4. In Figure 1, please indicate the number of records found in each database:

Records identified from:

Databases (n = 1749):

- IEEE Xplore (n = )

- ACM Digital Library (n = )

- Scopus (n = )

- Web of Science (n = )

5. Please present “Data extraction form” in the form of a table.

Yours sincerely,

Reviewer

Reviewer #3: This mini systematic review investigates the classification of anxiety in virtual reality (VR) using biosensors. The study aims to assess the evidence on the accurate classification of anxiety in VR. The review highlights the potential of psychophysiological and behavioral measures, such as heart rate, blood pressure, skin conductivity, pupil dilation, and breathing, for predicting anxiety levels. The authors provide a clear methodology for study inclusion and exclusion criteria and present their findings in a well-organized table. However, the review has limitations, including varying definitions of anxiety across studies and a lack of information on the use of various biosensors. The reviewer recommends referring to the DSM 5 for standardized definitions of stress and anxiety.

6. PLOS authors have the option to publish the peer review history of their article (what does this mean?). If published, this will include your full peer review and any attached files.

Reviewer #1: No

Reviewer #2: No

Reviewer #3: No

---

## [Author Response · Author response to Decision Letter 0]

26 Apr 2023

RESPONSE TO REVIEWERS

Reviewer 1

We thank the reviewer for their valuable comments. We tried to address the issues raised in the following steps:

The authors have not provided sufficient details of about the anxiety, VR and biosensors in the introduction part. They should expand their introduction providing more explanation of these terms which makes the basis of their work. Building on the knowledge of anxiety (or stress), VR and Biosensors, the importance of anxiety classification should be presented. The authors describes very little about its applications. More applications should be provided as evidence and their limitations should be included in the review.

As per this suggestion, we have expanded upon our introduction, adding the following sections:

Definition of Anxiety

Stress Response

Physiological and Behavioural Measures

We have also added some extra paragraphs into each section, including application areas.

The authors tried to establish their research questions but reading ahead it seems that they don’t answered their RQ and the review doesn’t look like a systemic approach. They should stick to their own RQs and provide detail answers in the light of published literature.

We have reevaluated the research questions to address this problem, as well as changed the methods into a scoping review as per another reviewer’s suggestion. Due to the change from systematic review to scoping review, we have come up with some changes to better investigate the literature.

Inlusion/exculsion criteria should be presented in a simple table.

Agreed, added Table 1.

No details of the classification’s methods are provided. Mentioning only the name classifiers does not provide any useful information to the researchers in the field.

We have expanded upon this section including some more information such as data preparation, feature extraction and selection, time windows and validation methods. Due to different types of measures, there is a lot of heterogeneity in this area and the information was kept brief to provide insight into current methods in the literature.

The discussion is just a summery of what they have presented earlier and not a real discussion over the problem of topic.

We have improved upon the discussion and added the following subsections:

Limitations

Future directions

We believe that these changes have improved the quality of our paper significantly. We are happy to make further adjustments if you have any other concerns.

Reviewer 2

Thank you very much for all your valuable insight, especially into PRISMA and systematic reviews. We tried to address your suggestions in the following steps:

1.In general, systematic reviews (SR) are intended to summarize knowledge on a particular topic, to provide specific practical guidelines for clinicians, and recommendations for future research. It is important that such conclusions can be drawn in relation to a specific population.

Thank you very much for your comment. We agree that a systematic review of the topic may not be the most suitable, especially considering how new it is. We have changed the submission into a scoping review.

2. The research question is usually very precise and allows to answer a specific clinical problem. Most of the times, a systematic review decides on the effectiveness of a therapy or compares different approaches to therapy. Meanwhile, the research questions posed by the Authors are precise; there are many of them, but they do not resolve the clinical problem.

We have changed the review type to scoping and adjusted the research questions accordingly. 

3. One of the most important elements of SR is the creation of the PICO and the inclusion and exclusion criteria. These elements are intended to ensure precision throughout the literature review process.

We have avoided using PICO because it is difficult to apply to computer science studies. We believe that changing the review type has made our questions more relevant. We also changed the inclusion and exclusion criteria into a table for enhanced comprehension.

4. SR requires a very precise approach. The guidelines clearly show the need to register the protocol a priori, or the participation of two independent Reviewers during the screening of abstracts, screening of full texts, and the qualitative assessment of the included articles. All these assumptions are intended to increase the relevance, reliability and objectivity of the review.

We agree with the reviewer on all accounts. The current submission is a mini-review, intended to inform on a small area of research, evaluate current progress and identify gaps in the literature. We do not believe a single reviewer would be enough for a detailed clinical systematic review or a meta-analysis. However, we do believe that it is useful for a quick scan of the literature to inform future work.

In conclusion, the Reviewer greatly appreciates the effort put into the review by the authors. Following the steps of the CONSORT guidelines is very engaging and demanding. The recommendations are as follows:

a) Change the name of the study design to “literature review” or “scoping review”; however, the Reviewer believes that "scoping review" would be a better option, as the study is based on many research questions and has an exploratory nature.

Agreed and changed.

b) It is recommended to delete the PRISMA checklist, the Authors should replace it with the PRISMA-ScR checklist.

Agreed, removed previous PRISMA and added File 2, PRISMA-ScR checklist.

Thank you very much for your insight on systematic and scoping reviews. Structured reviews are less common in computer science than in health, and we are very grateful to have an expert weigh in on the appropriate methods. We believe that the study is a lot more appropriately structured and labelled as a scoping review. We are grateful for your comments.

Reviewer 3

The authors mentioned in the beginning in the article that these two terms will be used interchangeably. However, this is not supported by the psychology literature, which clearly differentiates these terms. The author is pointed to the DSM 5 for detailed operationalizations of these terms.

We agree that the wording of this section gives the wrong impression. We have not intended to argue that stress and anxiety are the same concept. What we tried to convey was that despite there being a clear difference that needs to be acknowledged, the methods used for detecting each of these states share very similar traits, therefore it is useful to refer to them both during our review. We added a section called “definition of anxiety”, where we discuss this a bit more.

Citations were relevant, with most being from the past 10 years. However, there

were still some citations that seemed to be out-of-date. Unless these are seminal articles, it is recommended that the author update these citations.

We have added more recent citations into the introduction where the previous citations were outdated. Most of these citations are related to the use of VR in psychotherapy.

This table results answer the RQs; however, in the results section overall, it would be better to have a separate section based on each RQ because it was difficult to locate whether or not all the RQs were properly answered. Again, labeling sections using RQs is recommended.

We have reorganised the results section to better account for each question. Note that the questions have also been changed to better reflect the study more appropriately being titled a scoping review based on the suggestion of another reviewer.

There were significant limitations in the study starting from finding a definition of anxiety itself as different studies shows different definitions. However, the author is again referred to the DSM 5 for a standardized operationalization of the construct.

Thank you for this valuable suggestion. Hopefully, we addressed these issues a little better and discussed them more clearly. 

We believe that your suggestions have led to the betterment of our manuscript. We are very grateful for your time and suggestions.

---

## [Decision Letter · Decision Letter 1]

19 Jun 2023

Anxiety Classification in Virtual Reality Using Biosensors: A Mini Scoping Review

PONE-D-23-04739R1

Dear Dr. Mevlevioğlu,

We’re pleased to inform you that your manuscript has been judged scientifically suitable for publication and will be formally accepted for publication once it meets all outstanding technical requirements.

Kind regards,

Md. Milon Islam

Academic Editor

PLOS ONE

Additional Editor Comments (optional):

All the comments have been addressed properly.

Reviewers' comments:

Reviewer's Responses to Questions

**Comments to the Author**

1. If the authors have adequately addressed your comments raised in a previous round of review and you feel that this manuscript is now acceptable for publication, you may indicate that here to bypass the “Comments to the Author” section, enter your conflict of interest statement in the “Confidential to Editor” section, and submit your "Accept" recommendation.

Reviewer #2: All comments have been addressed

2. Is the manuscript technically sound, and do the data support the conclusions?

Reviewer #2: Yes

3. Has the statistical analysis been performed appropriately and rigorously? 

Reviewer #2: N/A

4. Have the authors made all data underlying the findings in their manuscript fully available?

Reviewer #2: Yes

5. Is the manuscript presented in an intelligible fashion and written in standard English?

Reviewer #2: Yes

6. Review Comments to the Author

Reviewer #2: Dear Authors,

thank you for responding to my comments. I believe that the current version of the manuscript can be accepted. Thank you for your efforts to improve your work.

Yours sincerely,

Reviewer

7. PLOS authors have the option to publish the peer review history of their article (what does this mean?). If published, this will include your full peer review and any attached files.

Reviewer #2: **Yes: **Adam Wrzeciono

---

## [Editor Report · Acceptance letter]

29 Jun 2023

PONE-D-23-04739R1 

Anxiety Classification in Virtual Reality Using Biosensors: A Mini Scoping Review 

Dear Dr. Mevlevioğlu:

I'm pleased to inform you that your manuscript has been deemed suitable for publication in PLOS ONE. Congratulations! Your manuscript is now with our production department. 

Kind regards, 

on behalf of

Dr. Md. Milon Islam 

Academic Editor

PLOS ONE